# RECONSTRUCTION AS SEQUENCE FOR EFFICIENT UNIFIED UNSUPERVISED ANOMALY DETECTION

## ABSTRACT

Unsupervised anomaly detection is highly desirable in industrial manufacturing processes due to the rarity of anomalies in real-world scenarios. Recent research has been focused on developing a unified framework for achieving multi-class anomaly detection. However, existing advanced feature-reconstruction-based methods often suffer from a lack of sufficient contextual awareness, thereby compromising the quality of the reconstruction. To address this challenge, we introduce a novel Reconstruction as Sequence (RAS) framework, which enhances the contextual correspondence during feature reconstruction through a sequence modelling perspective. In particular, based on the transformer technique, we integrate a specialized RASFormer block into the RAS framework. This block enables the capture of spatial relationships among different image regions and enhances temporal dependencies throughout the reconstruction process. By incorporating the RASFormer block, our RAS method achieves superior contextual awareness capabilities, leading to exceptional performance and faster inference speed. Experimental results show that our proposed RAS method significantly outperforms competing methods while exhibiting a maximal improvement of 29% in inference throughput. These results indicate the best trade-off between effectiveness and efficiency, further demonstrating the superiority and practicality of our method.

## 1 INTRODUCTION

Anomaly detection (AD) aims to identify outliers or abnormal regions for an input image. It is widely used in various fields such as industrial manufacturing (Bergmann et al., 2019a), healthcare (Fernando et al., 2021), and surveillance (Xia et al., 2020). Developing optimal AD models is challenging due to the rarity of anomalies in real-world scenarios. Researchers have explored unsupervised learning paradigms for AD without requiring anomaly-specific data. Nonetheless, they often build separate models for each class, which becomes inefficient as the number of classes increases. To overcome this limitation, recent research focuses on developing a unified AD framework capable of achieving unsupervised multi-class anomaly detection.

Recent works follow a feature-reconstruction paradigm to distinguish anomaly regions. Specifically, given an image $I$, they try to reconstruct its visual features $f$ through an encoder-decoder framework $\theta$: $\min ||f - \theta(f + \epsilon)||$ where $\epsilon$ represents the disturbance noise. UniAD (You et al., 2022) first incorporates the transformer architecture to construct the feature reconstruction model, ending up with superior performance. Wang et al. (2023) propose a unilaterally aggregated contrastive learning to obtain the concentrated inlier distribution as well as the dispersive outlier distribution. Lin et al. (2023) reconstruct the latent feature with the auto-encoder and the geometric transformation scheme.

Despite the promising performance, we observe that the existing advanced feature-reconstruction method, *i.e.,* UniAD (You et al., 2022), easily encounters difficulties in accurately reconstructing the original image. To visually demonstrate this problem, we train an image decoder, denoted as $\Theta$[1], to generate a reconstructed image $I'$ based on its reconstructed features, *i.e.,* $I' = \Theta[\theta(f)]$. We then compare this reconstructed image $I'$ with the original image $I$. Upon examining Fig. 1, we can see that some crucial object details, such as edges and lighting, are not adequately captured by

---

[1] $\Theta$ is designed to over-fit the test distribution, allowing it to perfectly map the features into the RGB image. Hence, it can reflect the quality of the reconstructed features.

UniAD. This observation suggests that UniAD is not able to align the reconstructed features $\theta(\boldsymbol{f})$ well with the image feature $\boldsymbol{f}$. Consequently, this limitation may result in false positive predictions for anomaly detection and ultimately lead to inferior performance.

To gain a deeper understanding of the reconstruction process, we encourage each decoding step to reconstruct the original image and inspect their difference. As depicted in the images at the top of Fig. 1, we can see that the differences among the successively reconstructed images are minimal in UniAD. This suggests that the decoder fails to capture the intricate patterns already reconstructed by the preceding decoder, resulting in limited contextual awareness throughout

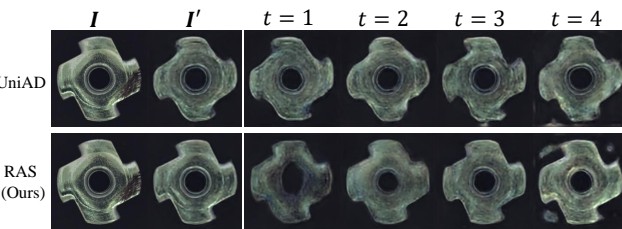

Figure 1: **Top:** inspection of the reconstruction failure of UniAD. **Bottom:** illustration of the superior reconstruction quality of our proposed RAS method. $\boldsymbol{I}$ is an anomalous metal nut.

the reconstruction process. This naturally raises a question: *can we enhance the contextual correspondence during the feature reconstruction?*

In this paper, we answer this question through a novel Reconstruction as Sequence (RAS) framework, which rethinks the feature reconstruction process from the perspective of sequence modelling for the unified unsupervised anomaly detection. Specifically, we consider each decoder layer as one step in the sequence model. Then, we specially design a RASFormer block which adapts the transformer architecture to the sequential feature reconstruction. Particularly, within the RASFormer block, we devise a novel strategy of adaptive gating to learn the contextual dynamics during feature reconstruction. By combining it with a transformer layer, our RASFormer block effectively captures spatial discrepancies and strengthens temporal dependencies, thereby enhancing the contextual awareness capability and achieving superior reconstruction quality (see Fig. 1).

To demonstrate the effectiveness of our RAS, we conduct comprehensive experiments and visualization analysis on two widely used benchmark datasets, *i.e.,* MVTec-AD (Bergmann et al., 2019a) and CIFAR-10 (Krizhevsky, 2009). Experimental results indicate that our proposed RAS can outperform competing methods while enjoying favorable throughput, leading to the best trade-off between the performance and efficiency. Quality visualization results demonstrate that the proposed RAS can well reconstruct images with high quality due to its effective enhancement of the contextual awareness capability. These results well demonstrate the effectiveness and superiority of our proposed RAS method.

Our contributions are summarized as follows:

- We thoroughly consider the contextual awareness capability during the feature reconstruction for the unified unsupervised AD. A novel Reconstruction as Sequence (RAS) method is proposed, which rethinks the feature reconstruction process from the sequence perspective.
- We introduce a generic RASFormer block to effectively enhance the contextual correspondence during the feature reconstruction, resulting in remarkable reconstruction outcomes.
- Experimental results validate the effectiveness of our method on two benchmark datasets, exhibiting a great trade-off between the performance and the efficiency.

## 2 RELATED WORK

**Unsupervised anomaly detection.** Due to the limited availability of anomalous samples, unsupervised learning methods are commonly employed in industrial quality inspection. Some works incorporate patch-level embedding (Yi & Yoon, 2020), geometric transformation (Golan & El-Yaniv, 2018), and elastic weight consolidation (Reiss et al., 2021), resulting in great improvement. Some works assume that a pre-trained network can extract discriminative features for anomaly detection (Defard et al., 2021; Rippel et al., 2021). For example, PaDiM (Defard et al., 2021) and MDND (Rippel et al., 2021) extract pre-trained features to model the normal distribution, subsequently utilizing a distance metric to identify anomalies. Nonetheless, these methods require memorizing all normal features, making them computationally expensive. Recent works are mostly based

on reconstruction-based anomaly detection, with an assumption that reconstruction models trained solely on normal samples perform well in normal regions but fail in anomalous regions (Bergmann et al., 2019b; Chen et al., 2022; Liu et al., 2020).

**Unified anomaly detection.** Many existing methods require training separate models for each class, which can become costly as the number of classes increases. Recently, the idea of constructing a unified model for multi-class anomaly detection has gained popularity in the research community. RegAD (Huang et al., 2022) addresses few-shot anomaly detection by training a single generalizable model, utilizing a limited number of normal images for each category during training. UniAD (You et al., 2022) employs a feature-reconstruction approach to pinpoint anomalous regions with the transformer architecture. OmniAL (Zhao, 2023) presents a panel-guided method to synthesize anomalies and achieve image reconstruction using dilated channel and spatial attention blocks.

Our RAS method shares a similar objective with UniAD (You et al., 2022) in identifying anomalies by aligning reconstructed features with the original image features. However, our method treats feature reconstruction as a sequence modeling problem and introduces the RASFormer block to enhance the contextual correspondence across different reconstruction steps. As a result, compared to UniAD, our RAS is more effective and efficient.

## 3 METHOD

### 3.1 PRELIMINARY

**Image feature extraction.** In the feature-reconstruction-based model, the goal is to align the reconstructed feature with the original image feature. To accomplish this, we follow previous method (You et al., 2022) by employing a pretrained convolutional neural network as the backbone for extracting the original image feature. This backbone is denoted as $\phi$, and the process of deriving features from the image $\boldsymbol{I}$ can be represented as $\{\boldsymbol{f}_1, ..., \boldsymbol{f}_n\} = \phi(\boldsymbol{I})$, where $n$ represents the number of feature levels. Consequently, for each feature level, we apply a $3 \times 3$ average pooling operation, resize them to the same size, and concatenate all the features along the channel dimension, yielding a comprehensive feature map:

$$\boldsymbol{f}_{\mathrm{org}} \in \mathbb{R}^{C_{\mathrm{org}} \times (H \times W)} = \mathrm{concat}\{\boldsymbol{f}_k | k = 1, .., n\} \tag{1}$$

where $C_{\mathrm{org}}$, $H$ and $W$ are the feature dimension, height and width of the feature map, respectively.

**Transformer layer.** Transformer (Vaswani et al., 2017; Devlin et al., 2018) has emerged as a foundational architecture in the field of artificial intelligence. A transformer layer comprises two essential sub-layers: the multi-head self-attention (MHSA) and the feed-forward network (FFN). To enhance training efficiency and performance, residual connections (He et al., 2016) and layer normalization (LN) (Ba et al., 2016) are applied to each sub-layer independently. Here, we utilize a post-LN transformer architecture (Xiong et al., 2020) to construct the transformer layer:

$$\mathrm{Transformer}(\boldsymbol{x}_q, \boldsymbol{x}_k, \boldsymbol{x}_v) = \mathrm{LN}(\mathrm{FFN}(\mathrm{LN}(\mathrm{MHSA}(\boldsymbol{W}_q \boldsymbol{x}_q, \boldsymbol{W}_k \boldsymbol{x}_k, \boldsymbol{W}_v \boldsymbol{x}_v)))) \tag{2}$$

where $\boldsymbol{x}_q$, $\boldsymbol{x}_k$, and $\boldsymbol{x}_v$ are input sequences of tokens. $\boldsymbol{W}_q$, $\boldsymbol{W}_k$, and $\boldsymbol{W}_v$ are all learnable parameters. For ease of description, we omit the residual connection in the above equation.

### 3.2 RECONSTRUCTION AS SEQUENCE (RAS)

#### 3.2.1 RETHINKING THE FEATURE RECONSTRUCTION FROM THE SEQUENCE PERSPECTIVE

**Denoised encoding.** The proposed RAS framework employs an encoder-decoder structure to reconstruct the image feature, *i.e.*, $\boldsymbol{f}_{\mathrm{org}}$, which is derived by a CNN backbone, as depicted in Eq. 1. The encoder is constructed by several transformer layers with feature jittering and neighbor masked attention mechanism as You et al. (2022):

$$\boldsymbol{f}' \in \mathbb{R}^{C_{\mathrm{rec}} \times (H \times W)} = \boldsymbol{W}_f(\boldsymbol{f}_{\mathrm{org}} + \boldsymbol{\epsilon}) \tag{3}$$

$$\boldsymbol{o}_e \in \mathbb{R}^{C_{\mathrm{rec}} \times (H \times W)} = \mathrm{Transformer}_{T_e}(...\mathrm{Transformer}_1(\boldsymbol{f}', \boldsymbol{f}', \boldsymbol{f}')) \tag{4}$$

where $\boldsymbol{W}_f \in \mathbb{R}^{C_{\mathrm{rec}} \times C_{\mathrm{org}}}$ and $C_{\mathrm{rec}}$ is the dimension of the latent reconstruction space. $T_e$ is the number of encoders. $\boldsymbol{\epsilon} = \{\boldsymbol{\epsilon}^i, i \in [0, H \times W)\}$ is the noisy features added to $\boldsymbol{f}_{\mathrm{org}}$ during training,

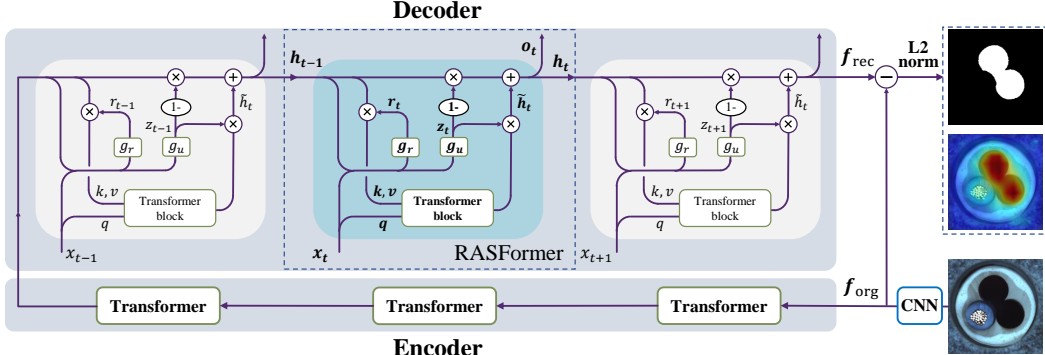

Figure 2: Overview of the proposed RAS framework, which follows an encoder-decoder architecture to perform feature reconstruction. We enhance the contextual awareness capability during feature reconstruction via a specially designed RASFormer block, denoted by the dotted box.

allowing the model to learn features of normal images through denoising:

$$\boldsymbol{\epsilon}^i \sim N(\mu = 0, \sigma^2 = (\alpha \frac{||\boldsymbol{f}_{\mathrm{org}}^i||_2}{C_{\mathrm{org}}})^2) \qquad (5)$$

where $\boldsymbol{f}_{\mathrm{org}}^i \in \mathbb{R}^{C_{\mathrm{org}}}$ is one element in $\boldsymbol{f}_{\mathrm{org}}$. $\alpha$ is the jittering scale to control the degree of noise. During the test phase, $\boldsymbol{\epsilon}$ is not applied following You et al. (2022).

**Sequence decoding.** UniAD (You et al., 2022) adopts conventional transformer layers to construct the decoder for feature reconstruction. Nonetheless, it is constrained in effectively capturing the contextual correspondence among decoding layers (see Fig. 1). In contrast, our proposed RAS framework considers the feature reconstruction process from a sequence perspective. Specifically, at each decoding step of $t$, we input an individual context embedding $\boldsymbol{x}_t \in \mathbb{R}^{C_{\mathrm{rec}} \times (H \times W)}$ and the previously reconstructed features $\boldsymbol{h}_{t-1}$ into the decoder $\theta_{\mathrm{dec}}^t$. $\theta_{\mathrm{dec}}^t$ performs feature aggregation and mapping from the latent reconstruction space into the image feature space:

$$\boldsymbol{h}_t \in \mathbb{R}^{C_{\mathrm{rec}} \times (H \times W)}, \boldsymbol{o}_t \in \mathbb{R}^{C_{\mathrm{org}} \times (H \times W)} = \theta_{\mathrm{dec}}^t(\boldsymbol{x}_t, \boldsymbol{h}_{t-1}) \qquad (6)$$

The aforementioned process can be repeated several times, resulting in a sequence of reconstructions. We randomly initialize $\boldsymbol{x}_t$ as previous works (You et al., 2022; Carion et al., 2020). $\boldsymbol{h}_{t-1}$ is the hidden state of previous decoder. We initialize the first hidden state with the output of the last encoder layer in Eq. 4, *i.e.*,$\boldsymbol{h}_0 = \boldsymbol{o}_e$. The reconstructed feature $\boldsymbol{f}_{\mathrm{rec}}$ can be denoted as $\boldsymbol{f}_{\mathrm{rec}} = \boldsymbol{o}_{T_d}$, where $T_e$ is the number of decoders. Noted that $\boldsymbol{x}_t$ can be considered the spatial version of image feature $\boldsymbol{f}_{\mathrm{org}}$ at the time step $t$, and $\boldsymbol{h}_t$ can be regarded as the temporal version after aggregating $\boldsymbol{x}_t$. The hidden states $\boldsymbol{h}_t$ and $\boldsymbol{h}_{t-1}$ are responsible for memorization of reconstruction knowledge, as in GRU (Cho et al., 2014).

To enhance the capability of contextual correspondence within the reconstruction sequence, we introduce a novel RASFormer block as the fundamental building block for decoders $\theta_{\mathrm{dec}}^t$. For ease of understanding, here we briefly represent the RASFormer block as a function, *i.e.*,$\theta_{\mathrm{dec}}^t = \mathrm{RASFormer}_t(\cdot)$. Details are provided in section 3.2.2.

### 3.2.2 RASFORMER BLOCK

The RASFormer block serves as a fundamental module in the decoder, playing a crucial role in capturing contextual correspondence within the sequential feature reconstruction process. We adhere to two guiding principles when designing the RASFormer block: 1) temporal dynamics, ensuring awareness of the previously captured information, alleviating the need to readdress it in subsequent reconstruction processes; 2) spatial dynamics, enabling the capture of relationships between the current input context embedding $\boldsymbol{x}_t$ and the previous knowledge $\boldsymbol{h}_{t-1}$ at the spatial level. To achieve this, we introduce a novel strategy of adaptive gating with transformers.

Specifically, given prior knowledge, *i.e.*,the previous hidden state $\boldsymbol{h}$, and the current input $\boldsymbol{x}$ (for ease of description, we leave out the subscript $t$), we first design a reset gate $\boldsymbol{r}_{\mathrm{reset}}$ to filter the prior

knowledge as follows:

$$r_{\text{reset}} = \sigma(W_r(h \oplus x)) \tag{7}$$

$$h_{\text{reset}} = r_{\text{reset}} \otimes h \tag{8}$$

where $\oplus$ is the concatenation of two feature maps along the channel dimension. $\otimes$ represents the element-wise multiplication between two matrices. $W_r$ is a learned weight matrix. $\sigma$ is a sigmoid function. Noted that all output tensors have the same shape as $h$, i.e., $\mathbb{R}^{C_{\text{rec}} \times (H \times W)}$.

We then incorporate the filtered knowledge $h_{\text{reset}}$ with the current input information $x$ through a transformer layer:

$$h_{\text{curr}} = \text{Transformer}(x, h_{\text{reset}}, h_{\text{reset}}) \tag{9}$$

With the reset gate $r_{\text{reset}}$, the current hidden state $h_{\text{curr}}$ can adaptively discard the previous hidden state $h$ and reset itself based on the remained prior knowledge and the current input $x$. As a result, prior knowledge that is deemed irrelevant in subsequent steps can be largely disregarded, leading to a more effective reconstruction process.

Moreover, we additionally utilize an update gate to refresh the reconstruction knowledge:

$$z_{\text{update}} = \sigma(W_z(h \oplus x)) \tag{10}$$

$$h_{\text{update}} = \text{LN}[(1 - z_{\text{update}}) \otimes h + z_{\text{update}} \otimes h_{\text{curr}}]. \tag{11}$$

where $\text{LN}(\cdot)$ means the layer normalization. The reconstruction output for the RASFormer block can be derived as follows:

$$o = W_o h_{\text{update}}. \tag{12}$$

The update gate $z_{\text{update}}$ governs the extent to which information from the previous hidden state $h$ is incorporated into the current hidden state $h_{\text{curr}}$. This enables our RASFormer to retain important contextual information, thereby enhancing the quality of the reconstruction output.

Summing it up, the RASFormer block can be summarized into a function:

$$h^*, o = \text{RASFormer}(x, h) \tag{13}$$

where $h^* = h_{\text{update}}$ is the updated hidden state.

**Remarks.** The reset gate can help filter out the previously reconstructed information, thus preventing wastage of the decoder's reconstruction capacity. The update gate enables the decoder to fully consider the discrepancy between the previously reconstructed information and the currently to-be-reconstructed information, thereby achieving remarkable reconstruction results. Additionally, the MHSA in the transformer layer can facilitate the effective interaction between each element in $x$ and other elements in $h$, enabling the capture of spatial dynamics. As a result, our RASFormer can effectively capture both temporal dependencies and spatial discrepancies, greatly enhancing the model's ability to establish contextual correspondence during feature reconstruction.

Our RASFormer block bears certain similarities to GRU (Cho et al., 2014), which is proposed for natural language processing. However, there are notable differences between these two methods. Unlike GRU, which operates on two-dimensional vector-like word embeddings, our RASFormer processes inputs with three-dimensional spatial feature maps. Besides, RASFormer further incorporates the transformer to explore the spatial relationships within the input feature maps, which is not managed in GRU.

### 3.2.3 LOSS AND INFERENCE

**Objective function.** The objective function for training RAS is to calculate the MSE loss between the original feature $f_{\text{org}}$ and the reconstructed feature $f_{\text{rec}}$.

$$\mathcal{L} = \frac{1}{H \times W} \|f_{\text{org}} - f_{\text{rec}}\|_2^2 \tag{14}$$

**Inference.** During the inference phase, the feature-level anomaly map $S_{\text{feat}}$ is computed by measuring the L2 norm of the difference between $f_{\text{org}}$ and $f_{\text{rec}}$.

$$S_{\text{feat}} = \|f_{\text{org}} - f_{\text{rec}}\|_2 \in \mathbb{R}^{H \times W} \tag{15}$$

The anomaly map is then up-sampled to the size of the original image using bi-linear interpolation to obtain the pixel-level anomaly map. The image-level anomaly score is derived by taking the maximum value of the averaged pooled pixel-level anomaly map.

Table 1: **Image-level AUROC for anomaly detection** on MVTec-AD (unified / separate).

| Category | | US | PaDiM | CutPaste | MKD | DRAEM | UniAD | RAS (ours) |
|---|---|---|---|---|---|---|---|---|
| Object | Bottle | 84.0 / 99.0 | 97.9 / 99.9 | 67.9 / 98.2 | 98.7 / 99.4 | 97.5 / 99.2 | 99.7/ 100 | **100** ± 0.00 / 100 |
| | Cable | 60.0 / 86.2 | 70.9 / 92.7 | 69.2 / 81.2 | 78.2 / 89.2 | 57.8 / 91.8 | 95.2/ 97.6 | **99.2** ± 0.12 / 99.7 |
| | Capsule | 57.6 / 86.1 | 73.4 / 91.3 | 63.0 / 98.2 | 68.3 / 80.5 | 65.3 / 98.5 | 86.9/ 85.3 | **92.6** ± 0.32 / 95.6 |
| | Hazelnut | 95.8 / 93.1 | 85.5 / 92.0 | 80.9 / 98.3 | 97.1 / 98.4 | 93.7 / 100 | 99.8 / 99.9 | **100** ± 0.00 / 100 |
| | Metal Nut | 62.7 / 82.0 | 88.0 / 98.7 | 60.0 / 99.9 | 64.9 / 73.6 | 72.8 / 98.7 | 99.2 / 99.0 | **99.9** ± 0.02 / 99.4 |
| | Pill | 56.1 / 87.9 | 68.8 / 93.3 | 71.4 / 94.9 | 79.7 / 82.7 | 82.2 / 98.9 | 93.7 / 88.3 | **96.3** ± 0.35 / 96.2 |
| | Screw | 66.9 / 54.9 | 56.9 / 85.8 | 85.2 / 88.7 | 75.6 / 83.3 | 92.0 / 93.9 | 87.5/ 91.9 | 95.3 ± 0.40 / 95.6 |
| | Toothbrush | 57.8 / 95.3 | 95.3 / 96.1 | 63.9 / 99.4 | 75.3 / 92.2 | 90.6 / 100 | 94.2/ 95.0 | **98.7** ± 0.30 / 94.8 |
| | Transistor | 61.0 / 81.8 | 86.6 / 97.4 | 57.9 / 96.1 | 73.4 / 85.6 | 74.8 / 93.1 | **99.8**/ 100 | 99.2 ± 0.00 / 100 |
| | Zipper | 78.6 / 91.9 | 79.7 / 90.3 | 93.5 / 99.9 | 87.4 / 93.2 | **98.8** / 100 | 95.8 / 96.7 | 98.4 ± 0.07 99.4 |
| Texture | Carpet | 86.6 / 91.6 | 93.8 / 99.8 | 93.6 / 93.9 | 69.8 / 79.3 | 98.0 / 97.0 | **99.8** / 99.9 | 99.5 ± 0.05 / 100 |
| | Grid | 69.2 / 81.0 | 73.9 / 96.7 | 93.2 / 100 | 83.8 / 78.0 | 99.3 / 99.9 | 98.2/ 98.5 | **99.8** ± 0.16 / 100 |
| | Leather | 97.2 / 88.2 | 99.9 / 100 | 93.4 / 100 | 93.6 / 95.1 | 98.7 / 100 | **100** / 100 | **100** ± 0.00 / 100 |
| | Tile | 93.7 / 99.1 | 93.3 / 98.1 | 88.6 / 94.6 | 89.5 / 91.6 | 99.8 / 99.6 | 99.3/ 99.0 | **100** ± 0.02 / 99.9 |
| | Wood | 90.6 / 97.7 | 98.4 / 99.2 | 80.4 / 99.1 | 93.4 / 94.3 | **99.8** / 99.1 | 98.6/ 97.9 | 98.7 ± 0.23 / 98.5 |
| Mean | | 74.5 / 87.7 | 84.2 / 95.5 | 77.5 / 96.1 | 81.9 / 87.8 | 88.1 / 98.0 | 96.5 / 96.6 | **98.4** ± 0.08 / 98.6 |

# 4 EXPERIMENTS

## 4.1 EXPERIMENT SETUPS

**MVTec-AD.** MVTec-AD is a widely used benchmark for image anomaly detection, including 15 categories of industrial products and defects. It consists of 3,629 anomaly-free images for training and 1,725 images for the test set. For the test set, both normal and anomalous samples are provided.

**CIFAR-10.** CIFAR-10 is an image classification dataset consisting of 60,000 images of 32×32 pixels, distributed across 10 classes with 6,000 images per class. Following previous work (You et al., 2022), we employ a many-versus-many setting, where half of the classes are used as normal samples during training, while the remaining half are used as anomalies during testing.

**Evaluation Metrics.** The performance of anomaly detection models is typically measured by AU-ROC. Following previous works (You et al., 2022; Defard et al., 2021; Chen et al., 2022), we report the image-level AUROC and the pixel-level AUROC on MVTec-AD. On CIFAR-10, model performance is evaluated using image-level AUROC.

**Implementation details.** For a fair comparison, we use EfficientNet-B4 as the backbone following UniAD (You et al., 2022). Images are resized to $224 \times 224$ before input into the backbone. Feature maps of the image are extracted from levels 1 to 4. We align them to the size of their highest-level feature map, $i.e., 14 \times 14$. We utilize the AdamW optimizer with a learning rate of $7e - 4$ and a weight decay of $1e - 4$. The batch size is set to 64. All models are trained with 500 epochs.

## 4.2 COMPARISON WITH STATE-OF-THE-ART METHODS

In this section, we present a comparison between our proposed RAS method and existing methods. Specifically, for the one-class competing models, we directly leverage the baseline models reported in UniAD. As for the unified AD baseline methods, we focus on ensuring a fair comparison with feature-reconstruction methods[2].

**Performance comparison on MVTec-AD**. We compare our method with several classical methods under both the unified and separate settings. We report the performance at the image level and pixel level on MVTec-AD in Table 1 and Table 2. Among these baseline methods, UniAD is specifically designed for the unified setting, while the others follow the separate setting. We can see that, for the *unified* unsupervised anomaly detection, our method can outperform UniAD with a significant improvement of 1.9% AUROC for image-level anomaly detection and of 0.7% AUROC for pixel-level

---

[2]OmniAL (Zhao, 2023) introduces pseudo-synthesized anomalies and utilizes a computationally expensive image segmentation framework, and is thus ignored for a fair comparison.

Table 2: **Pixel-level AUROC for anomaly localization** on MVTec-AD (unified / separate).

| | Category | US | PaDiM | FCDD | MKD | DRAEM | UniAD | RAS (ours) |
|---|---|---|---|---|---|---|---|---|
| Object | Bottle | 67.9 / 97.8 | 96.1 / 98.2 | 56.0 / 97 | 91.8 / 96.3 | 87.6 / 99.1 | 98.1 / 98.1 | **98.4** ± 0.02 / 98.5 |
| | Cable | 78.3 / 91.9 | 81.0 / 96.7 | 64.1 / 90 | 89.3 / 82.4 | 71.3 / 94.7 | 97.3 / 96.8 | **98.7** ± 0.03 / 98.6 |
| | Capsule | 85.5 / 96.8 | 96.9 / 98.6 | 67.6 / 93 | 88.3 / 95.9 | 50.5 / 94.3 | 98.5 / 97.9 | **98.6** ± 0.01 / 98.6 |
| | Hazelnut | 93.7 / 98.2 | 96.3 / 98.1 | 79.3 / 95 | 91.2 / 94.6 | 96.9 / 99.7 | 98.1 / 98.8 | **98.5** ± 0.02 / 98.7 |
| | Metal Nut | 76.6 / 97.2 | 84.8 / 97.3 | 57.5 / 94 | 64.2 / 86.4 | 62.2 / 99.5 | 94.8 / 95.7 | **97.3** ± 0.12 / 98.1 |
| | Pill | 80.3 / 96.5 | 87.7 / 95.7 | 65.9 / 81 | 69.7 / 89.6 | 94.4 / 97.6 | 95.0 / 95.1 | **98.3** ± 0.11 / 98.2 |
| | Screw | 90.8 / 97.4 | 94.1 / 98.4 | 67.2 / 86 | 92.1 / 96.0 | 95.5 / 97.6 | 98.3 / 97.4 | **99.1** ± 0.03 / 99.1 |
| | Toothbrush | 86.9 / 97.9 | 95.6 / 98.8 | 60.8 / 94 | 88.9 / 96.1 | 97.7 / 98.1 | **98.4** / 97.8 | **98.4** ± 0.01 / 98.5 |
| | Transistor | 68.3 / 73.7 | 92.3 / 97.6 | 54.2 / 88 | 71.7 / 76.5 | 64.5 / 90.9 | 97.9 / 98.7 | **98.9** ± 0.03 / 99.1 |
| | Zipper | 84.2 / 95.6 | 94.8 / 98.4 | 63.0 / 92 | 86.1 / 93.9 | **98.3** / 98.8 | 96.8 / 96.0 | 97.8 ± 0.03 / 97.7 |
| Texture | Carpet | 88.7 / 93.5 | 97.6 / 99.0 | 68.6 / 96 | 95.5 / 95.6 | **98.6** / 95.5 | 98.5/ 98.0 | 97.9 ± 0.07 / 98.7 |
| | Grid | 64.5 / 89.9 | 71.0 / 97.1 | 65.8 / 91 | 82.3 / 91.8 | **98.7** / 99.7 | 96.5 / 94.6 | 97.1 ± 0.03 / 97.2 |
| | Leather | 95.4 / 97.8 | 84.8 / 99.0 | 66.3 / 98 | 96.7 / 98.1 | 97.3 / 98.6 | **98.8** / 98.3 | 98.7 ± 0.05 / 99.2 |
| | Tile | 82.7 / 92.5 | 80.5 / 94.1 | 59.3 / 91 | 85.3 / 82.8 | **98.0** / 99.2 | 91.8 / 91.8 | 92.9 ± 0.14 / 94.1 |
| | Wood | 83.3 / 92.1 | 89.1 / 94.1 | 53.3 / 88 | 80.5 / 84.8 | **96.0** / 96.4 | 93.2 / 93.4 | 92.0 ± 0.23 / 92.9 |
| Mean | | 81.8 / 93.9 | 89.5 / 97.4 | 63.3 / 92 | 84.9 / 90.7 | 87.2 / 97.3 | 96.8 / 96.6 | **97.5** ± 0.01 / 97.8 |

Table 3: **Image-level AUROC for anomaly detection** under the unified case on CIFAR-10.

| Normal Indices | US | FCDD | PANDA | MKD | UniAD | RAS (ours) |
|---|---|---|---|---|---|---|
| {01234} | 51.3 | 55.0 | 66.6 | 64.2 | 84.4 | **85.8** ± 0.14 |
| {56789} | 51.3 | 50.3 | 73.2 | 69.3 | 80.9 | **84.1** ± 0.04 |
| {02468} | 63.9 | 59.2 | 77.1 | 76.4 | 93.0 | **93.4** ± 0.03 |
| {13579} | 56.8 | 58.5 | 72.9 | 78.7 | 90.6 | **92.4** ± 0.08 |
| Mean | 55.9 | 55.8 | 72.4 | 72.1 | 87.2 | **88.9** ± 0.03 |

anomaly localization. Although our RAS is not specifically designed for the conventional *separate* setting, it achieves comparable performance to conventional advanced methods, *e.g.,*DRAEM (Zavrtanik et al., 2021). Compared to UniAD, our method can also obtain a 2.0% improvement in image-level AUROC and a 1.2% increase in terms of the pixel-level AUROC in the seperate setting.

**Performance comparison on CIFAR-10**. Following You et al. (2022), we select four different combinations of classes for training and testing. In Table 3, normal indices {0,1,2,3,4} means that classes with indices 0-4 are used as normal samples in the training set, while classes with indices 5-9 are treated as anomalies during testing. Similar explanations apply for the other combinations. We can observe that the proposed RAS method can consistently outperform all competing methods, achieving the best results in each case. The average AUROC across the 4 combinations reaches 88.9%, which presents a 1.7% improvement over UniAD.

**Efficiency comparison**. We also conduct a comparison between our method and UniAD in terms of inference efficiency. As illustrated in Table 4, our RAS model significantly outperforms UniAD with higher throughput.

Table 4: **Efficiency comparison** between UniAD and RAS.

| model | $T_e$ - $T_d$ | Performance | Throughput | Improvement |
|---|---|---|---|---|
| UniAD | 4 - 4 | 96.5 / 96.8 | 115 fps | - |
| RAS | 0 - 1 | 96.5 / 97.2 | 148 fps | 29% |
| | 2 - 4 | 98.4 / 97.5 | 125 fps | 9% |

Specifically, when equipped with only a single decoder layer, RAS achieves better performance than UniAD while exhibiting a remarkable throughput with an improvement of 29%. Moreover, when the number of encoders and decoders is increased to 2 and 4 respectively, RAS can obtain an improvement of 9% in terms of the inference throughput, while maintaining notable improvements of 1.9% and 0.7% in image-level and pixel-level performance. These results strongly demonstrate the favorable efficiency of our proposed method compared to the baseline model, highlighting the potential for practical applications where computational resources are constrained.

**Summary**. Overall, our proposed RAS method exhibits substantial superiority over competing methods on both benchmark datasets, with a noticeable margin. In addition to its superior perfor-

Table 5: **Component analysis** from the macro and micro aspects.

(a) Macro: number of encoder and decoder

|  | $T_d = 1$ | $T_d = 2$ | $T_d = 3$ | $T_d = 4$ |
|---|---|---|---|---|
| $T_e = 0$ | 96.5 / 97.2 | 97.7 / 97.3 | 97.9 / 97.3 | 98.0 / 97.4 |
| $T_e = 1$ | 97.1 / 97.3 | 97.8 / 97.4 | 98.2 / 97.5 | 98.3 / 97.4 |
| $T_e = 2$ | 97.6 / 97.4 | 98.0 / 97.5 | 98.2 / 97.5 | **98.4 / 97.5** |

(b) Micro: gate and transformer

| gate | transformer | AUROC |
|---|---|---|
| ✓ |  | 97.5 / 97.3 |
|  | ✓ | 96.1 / 97.2 |
| ✓ | ✓ | **98.4 / 97.5** |

Table 6: **Robustness analysis** on the feature reconstruction. Values in parentheses is the difference compared to the result of jittering scale of 0.

| jittering scale, *i.e.*,$\alpha$ |  | 0 | 10 | 20 | 30 | 40 | 50 |
|---|---|---|---|---|---|---|---|
| image-level | UniAD | 96.5 | 96.4 (-0.1) | 96.2 (-0.3) | 95.1 (-1.4) | 90.9 (-5.6) | 83.7 (-12.8) |
|  | RAS | **98.4** | 98.4 (**-0.0**) | 98.2 (**-0.2**) | 97.6 (**-0.8**) | 96.6 (**-1.8**) | 91.9 (**-6.5**) |
| pixel-level | UniAD | 96.8 | 96.8 (-0.0) | 96.7 (-0.1) | 96.5 (-0.3) | 95.5 (-1.3) | 92.0 (-4.8) |
|  | RAS | **97.5** | 97.5 (**-0.0**) | 97.5 (**-0.0**) | 97.4 (**-0.1**) | 97.2 (**-0.3**) | 96.4 (**-1.1**) |

mance, RAS also shows favorable efficiency with a remarkable inference throughput. These results well demonstrate the effectiveness and superiority of our proposed RAS method.

### 4.3 MODEL ANALYSIS

**Component analysis.** We investigate the impact of components in the proposed RAS framework from two aspects. (1) Macro: we investigate the performance of RAS with different numbers of encoder-decoder pairs in Table 5(a). We can observe that increasing the number of encoder layers or decoder layers can bring a substantial performance improvement. However, as the layer number increases, the performance gradually saturates. (2) Micro: we analyze the impact of the adaptive gating strategy and the transformer in the RASFormer block in Table 5(b). We can observe that no gate or no transformer can severely degrade the model performance, indicating their positive roles. It can also demonstrate the benefit of capturing the spatial dynamics and the temporal dynamics during the feature reconstruction.

**Training convergence.** We present a visualization of the changes in the loss function and performance during the entire training process. As depicted in Fig. 3, we can observe that our RAS consistently achieves lower loss values compared to UniAD. This demonstrates the remarkable feature reconstruction capability of RAS for unified unsupervised anomaly detection. Furthermore, our RAS exhibits rapid convergence to high performance, indicating its ability to effectively utilize limited training data and significantly improve training efficiency.

**Robustness of reconstruction**. In this experiment, we evaluate the robustness of the reconstruction capability of the anomaly detection model when exposed to noisy test data. To achieve this, we introduce various levels of per-

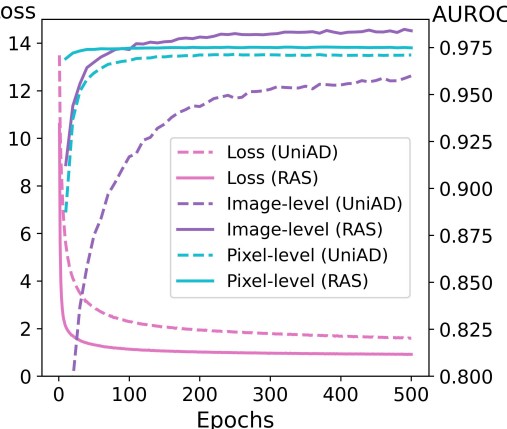

Figure 3: **Loss and performance** during training.

turbed noise into the input image by adjusting the jittering scale factor, *i.e.*,$\alpha$ in Eq. 5, and analyze the resulting changes in performance. The experimental results are presented in Table 6. We observe that when $\alpha = 10$, RAS performs nearly on par with the condition where no feature jitter is applied, while UniAD experiences a slight 0.1% decrease in image-level AUROC. However, as we increase the scale of noise starting from 20, the performance gap between the two models gradually widens. Notably, when subjected to strong noise at $\alpha = 50$, UniAD's performance degradation exceeds that of RAS by more than 4.0× in image-level AUROC and approximately 2.0× in pixel-level AUROC. These results effectively demonstrate the robustness of our proposed method, highlighting its ability to handle noisy test data and maintain superior performance under challenging conditions.

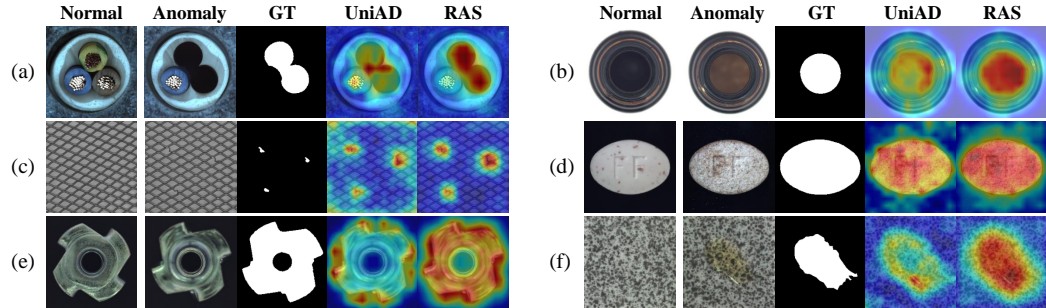

Figure 4: **Qualitative results for anomaly localization** on MVTec-AD. We turn the anomaly map into the heat map for better visualization. Regions with higher anomaly scores are depicted in vibrant red colors. Best viewed in colors.

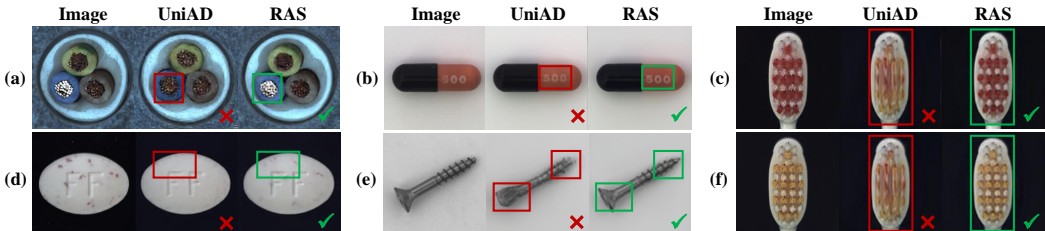

Figure 5: **Visualization comparison of image reconstruction** between UniAD and RAS. We utilize bounding boxes to visually differentiate between the worse (red) and better (green) regions.

## 4.4 QUALITATIVE RESULTS

**Anomaly map.** To better reveal the advantage of our proposed RAS model, we also conduct a qualitative investigation of the anomaly maps generated by UniAD and our RAS. As shown in Fig. 4, it is evident that our RAS can better localize the anomaly regions. For instance, in examples (a), (c), and (e), our proposed method generates more accurate anomaly maps compared to UniAD. Moreover, in examples (b), (d), and (f), compared to UniAD, our RAS successfully emphasize the salience of anomalous regions by yielding higher anomaly scores. These qualitative findings effectively demonstrate the benefits of enhancing the contextual awareness capability during feature reconstruction, highlighting the superiority of our proposed RAS model.

**Quality of image reconstruction.** The superiority of our method is not only evident in the anomaly maps but also reflected in the detailed image reconstruction. Fig. 5 presents a side-by-side comparison of the reconstructed images generated by RAS and UniAD. It is clearly observed that RAS provides more accurate reconstruction of image details. For example, in (a), RAS accurately reproduces the reflection of the cable wire in the left-bottom area. In (e), RAS correctly replicates the head and tail of the screw, while UniAD fails to do so. These results demonstrate that the reconstructed features in RAS are more aligned with the ground truth, resulting in superior image reconstruction and anomaly detection performance.

## 5 CONCLUSION

In this paper, we propose a novel Reconstruction as Sequence (RAS) framework for the unified unsupervised anomaly detection. RAS rethinks the feature reconstruction from the sequence perspective. Therefore, a generic RASFormer block is designed to capture the contextual dependencies among the sequential reconstruction. Thanks to the RASFormer block, our RAS can enhance the contextual awareness capability during feature reconstruction, leading to superior performance. Experimental results on standard benchmark datasets show that the proposed RAS can consistently outperform competing methods with a notable margin. Additionally, efficiency analysis demonstrate that the proposed RAS can achieve remarkable inference throughput compared with baseline methods as well. These results well demonstrate the effectiveness and superiority of the proposed method.

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
