# OpenReview forum: "Reconstruction as Sequence for Efficient Unified Unsupervised Anomaly Detection"
_ICLR.cc/2024/Conference — ICLR 2024 Conference Withdrawn Submission_

### Official Review · Reviewer_xekD · 2023-10-15

**Soundness:** 3 good
**Presentation:** 3 good
**Contribution:** 2 fair
**Rating:** 3
**Confidence:** 5

**Summary:**

The authors study unsupervised anomaly detection in images, presenting a reconstruction-based framework that combines transformers and GRUs to improve the fidelity of the reconstruction (decoding) stage. The proposed method is compared with a set of baselines from literature on two datasets: MVTec AD and CIFAR-10.

**Strengths:**

- The paper is well organized.
- The authors address an actively studied field.

**Weaknesses:**

- The method is not compared with recent state-of-the-art methods reporting better results (see [1]). The authors claim to achieve state-of-the-art results, but according to [1], this is clearly not the case. Since the method does not surpasses competing methods, its benefits are not well justified from a practical point of view.
- CIFAR-10 is rather consider as a toy dataset, while MVTec AD is rather small in size. The proposed method should be evaluated on additional datasets to demonstrate its generalization capacity.
- The proposed method is constructed on top of UniAD, which represents an incremental / limited contribution. The RAS framework should be demonstrate in conjunction with multiple approaches to be evidenced as a standalone contribution.
- The ablation study is not comprehensive enough. For example, the contribution / necessity of each type of gate is not shown.
- The implementation details are insufficient to reproduce the results. For example, the number of attention heads is not mentioned.

[1] https://paperswithcode.com/sota/anomaly-detection-on-mvtec-ad

**Questions:**

The are no particular questions besides addressing the weaknesses.

---

### Official Review · Reviewer_m5gg · 2023-10-23

**Soundness:** 3 good
**Presentation:** 3 good
**Contribution:** 3 good
**Rating:** 6
**Confidence:** 4

**Summary:**

This paper proposes an image-based anomaly detection (both at image-level and pixel-level evaluations) method based on the reconstruction of the data with a framework having so-called RASFormer, which enables capturing the spatial relationships among different image regions by enabling the "temporal" dependencies during the reconstruction process. The experimental analysis was applied to two standard datasets with an in-depth analysis of the method as well as comparing it with the SOTA.

**Strengths:**

- Although the proposed method's RASFormer is very similar to GRU, it is somehow distinguished from GRU by incorporating a transformer for exploring spatial information within the feature maps.
- The experimental analysis demonstrates that the proposed method brings in improvements, particularly for the cases where contextual correspondence is important (such as industrial manufacturing).

**Weaknesses:**

1) Overall, the paper's writing is satisfactory; however, the introduction/abstract should be improved. For instance, the first sentence makes a very strong claim, even though in some cases, the anomaly detection task can classify an image as normal or abnormal without necessarily localizing the abnormal regions. Additionally, the last sentence of the introduction needs proper citations. Moreover, I fail to see the significance of specifically mentioning "industrial manufacturing" in the opening sentence.

2) It is unclear whether the metrics used in this paper align with prior research. For example, in a recent ICCV work [A], the reported results of prior research do not always match those presented in this paper.
[A] https://openaccess.thecvf.com/content/ICCV2023/papers/Yao_Focus_the_Discrepancy_Intra-_and_Inter-Correlation_Learning_for_Image_Anomaly_ICCV_2023_paper.pdf

3) Several related methods exist, especially in the proceedings of ICCV 2023, which the authors may not have been able to include in their comparisons. However, it is important to note that some of these methods, in certain cases, outperform the proposed approach and should be considered for inclusion. Nevertheless, I do not consider this a reason for rejection, given the constraints of ICCV data and the submission deadline of ICLR and particularly due to fact that the paper brings in technical novelties to the task.

**Questions:**

- Weakness (2)
Please also comment on (3)

---

### Official Review · Reviewer_5D2p · 2023-10-30

**Soundness:** 3 good
**Presentation:** 2 fair
**Contribution:** 2 fair
**Rating:** 5
**Confidence:** 4

**Summary:**

This work proposes a method for unsupervised anomaly detection. Basically, it adopts the framework of a previous method (UniAD) and improves UniAD by proposing a RASFormer block to replace the vanilla Transformer block. The motivation of RASFormer is to enhance the contextual correspondence between decoded features and is implemented by using a reset gate to filter the prior decoded features. On the pixel-level task and image-level task, the proposed method (RAS) outperforms UniAD and other anomaly detection methods.

**Strengths:**

1. The proposed RASFormer block is computationally efficient and easy to implement.
2. RASFormer experimentally improves the original transformer block.
3. The experiments are extensive and convincing.

**Weaknesses:**

1. Writing
The role of contextual embedding x_t should be carefully explained. Why x_t is a 'spatial version of image feature'?  How is x_t updated?
The authors claim the 'reconstruction as sequence (RAS)' framework is a key contribution of the method. However, it seems the main framework of RAS is the same as UniAD.

2. Novelty
The framework of RAS is similar to UniAD. The gating method in RASFormer block looks like GRU or LSTM. Although RAS outperforms UniAD experimentally, the novelty and overall contribution are limited.

**Questions:**

None.

---

### Official Review · Reviewer_rUeR · 2023-10-31

**Soundness:** 2 fair
**Presentation:** 3 good
**Contribution:** 1 poor
**Rating:** 3
**Confidence:** 4

**Summary:**

An anomaly detection process is proposed in the industrial manufacturing processes. The proposed method is based on reconstructing an input image (or its features) using an auto-encoder type transformer based deep network which is trained on the normal data only. For inference, if the reconstruction error is small, the input is considered as normal, otherwise  anomalous.  A Reconstruction as Sequence (RAS) framework is proposed  which reconstructs the image in multiple steps or stages and tries to better utilise the contextual information available within the image. Results are demonstrated on an often used MVTec-AD dataset and also on CIFAR10 dataset considering some classes as normal and training the model using only these classes. The remaining classes are considered as abnormal and later the trained model tries to identify a given image as normal or abnormal. One may think of such algorithms as normal-supervised (or one-class supervision) instead of using the term `unsupervised'. It is because, if during training some data from abnormal classes is also included in the training data, the discrimination will be lost. Therefore these methods are indeed not `unsupervised' rather  have normal class supervison.

**Strengths:**

1. A transformer based architecture is proposed which better utilises the context and relationships among different parts of the input image.
2. Good performance is reported on two datasets.
3. The proposed method has outperformed UniAD method.

**Weaknesses:**

1. The usage of terms temporal dependencies, temporal dynamics is confusing since the proposed algorithm handles only images, not videos.

2. Authors have focussed too much on a NIPS2022 paper UniAD, while many good papers have appeared after that work such as: PNI : Industrial Anomaly Detection using Position and Neighbourhood Information (ICCV23) having  99.56 (image level) 98.98 (pixel level) performance.

3. Performance improvement compared to the baseline UniAD  is just 1.9% and 0.7% on MVTec AD. Such improvement  cannot be considered as significant.

4. For throughput comparisons, much faster methods are also available.  See the papers with code `Anomaly Detection on MVTec AD' webpage.

5. Most SOTA methods are reporting results on additional anomaly detection datasets like BTAD, VisA etc, where as CIFAR10 although employed by UNiAD, is a classification dataset. For a better comparison with other methods, same set of datasets should have been used.

6. The compared methods US, CutPaste, MKD, FCDD are neither introduced nor referred. It is perhaps due to copying results from UniAD without even referring who are the compared methods.

7. In Table 4, 29% more throughput is obtained when no encoder transformer is used while a single decoder transformer is employed (Te=0, Td=1). That would mean the CNN features are directly input to decoder transformer  as one can infer from Fig. 2. Decoder will change those features to image? I guess there is no sequence or step by step decoding in this case. Authors need to explain how decoding is done without encoding step?

**Questions:**

Please answer all questions in weakness section.